# Ensemble Docking as a Tool for the Rational Design of Peptidomimetic *Staphylococcus aureus* Sortase A Inhibitors

**DOI:** 10.3390/ijms252011279

**Published:** 2024-10-20

**Authors:** Dmitry A. Shulga, Konstantin V. Kudryavtsev

**Affiliations:** 1Department of Chemistry, Lomonosov Moscow State University, Leninskie Gory 1/3, 119991 Moscow, Russia; 2Vreden National Medical Research Center of Traumatology and Orthopedics, 195427 St. Petersburg, Russia

**Keywords:** Sortase A, *Staphylococcus aureus*, multidrug resistance, antivirulence drugs, ensemble docking, protein dynamics, peptidomimetic, oligopeptide

## Abstract

Sortase A (SrtA) of *Staphylococcus aureus* has long been shown to be a relevant molecular target for antibacterial development. Moreover, the designed SrtA inhibitors act via the antivirulence mechanism, potentially causing less evolutional pressure and reduced antimicrobial resistance. However, no marketed drugs or even drug candidates have been reported until recently, despite numerous efforts in the field. SrtA has been shown to be a tough target for rational structure-based drug design (SBDD), which hampers the regular development of small-molecule inhibitors using the available arsenal of drug discovery tools. Recently, several oligopeptides resembling the sorting sequence LPxTG (Leu-Pro-Any-Thr-Gly) of the native substrates of SrtA were reported to be active in the micromolar range. Despite the good experimental design of those works, their molecular modeling parts are still not convincing enough to be used as a basis for a rational modification of peptidic inhibitors. In this work, we propose to use the ensemble docking approach, in which the relevant SrtA conformations are extracted from the molecular dynamics simulation of the LPRDA (Leu-Pro-Arg-Asp-Ala)-SrtA complex, to effectively represent the most significant and diverse target conformations. The developed protocol is shown to describe the known experimental data well and then is applied to a series of new peptidomimetic molecules resembling the active oligopeptide structures reported previously in order to prioritize structures from this work for further synthesis and activity testing. The proposed approach is compared to existing alternatives, and further directions for its development are outlined.

## 1. Introduction

*Staphylococcus aureus* (*S. aureus*) is responsible for many medical complications in patients, ranging from sinusitis to sepsis after joint surgery [1,2]. Despite several antibacterial agents being active against *S. aureus*, the latter have managed to develop drug resistance, resulting in Methicillin-resistant *Staphylococcus aureus* (MRSA), a significant problem for possible treatment [3,4,5]. It is believed that the main cause of such drug resistance is the evolutionary pressure caused by drugs that target the existence of the bacteria. Hence, to diminish the drug resistance development mechanism, it has been proposed to target the virulence of the bacteria, i.e., the routes that bacteria use to impact the host [6,7,8,9,10], instead of trying to kill it completely. In that respect, Sortase A (SrtA) of *Staphylococcus aureus*, a membrane-bound cysteine transpeptidase, which helps to display the bacterial outer shell proteins by specifically cleaving the “sorting sequence” LPxTG (Leu-Pro-Any-Thr-Gly) of those proteins, has long been established as a promising target for the discovery of antibacterial drugs. The structure and dynamics of SrtA have been extensively described previously [4,11,12,13,14]. The main benefits of SrtA as a drug target are:Its function is related only to bacterial cell adhesion to the host cells and immune system evasion, not bacterial survival, which diminishes the risks of developing resistant stem;The close analogs of the SrtA protein are absent in humans, which implies lower risks of potential off-target toxicity;SrtA is located in the external bacterial cell envelope, thus being much more readily accessible to potential drugs compared to internal bacterial cell targets;The pharmacological significance of the target was validated [15,16] using several models.

Significant efforts have been made to find proper hit compounds [9,17,18,19,20,21,22], which can be generally split into either substrate mimetics, natural compounds, or diverse small-molecule families [23]. Early attempts were mostly concentrated on covalent inhibitors (e.g., [24]), whereas the recent focus has mostly been on non-covalent inhibitors, because of the potentially lower off-target toxicity and, hence, wider the therapeutic window [4].

Despite all the attractiveness of the SrtA drug target [9,16,25], to date, not a single marketed drug or even an efficacious and developable drug candidate has been reported [10,26,27].

Such a striking contrast of this unmet need and the great number and diversity [28] of the hit molecules reported with the lack of lead compounds forming a series of structures with explainable structure–activity relationships (SARs) clearly posits a problem that requires a reasonable explanation. We hypothesized earlier [29] that the SrtA binding site represents a tough challenge to the simple means of current structure-based computer-aided drug discovery. Firstly, the site is rather extended and shallow, thus resembling the typical sites involved in protein–protein interactions (PPIs) [30,31]. Secondly, its two edges are represented by two highly flexible loops—β6-β7 and β7-β8 [11,12]. In depth, a molecular dynamics study of SrtA revealed that several distinct binding modes of the sorting sequence LPATG* (Leu-Pro-Ala-Thr-Gly*) are involved at different stages of the substrate binding, preceding the T-G (Thr-Gly) cleavage event [12]. On the one hand, the site flexibility at body temperature means that any putative and energetically achievable SrtA conformation can be used for successful structure-based drug design (SBDD). On the other hand, the number of achievable conformations is large, and many of them represent distinct arrangements of the features responsible for affine interactions of a ligand with the receptor. Moreover, there is a rational concern that not only structural features but also the dynamical coherence of the ligand–receptor complex might be necessary to ensure binding with significant affinity. Several applied studies using certain parts of SBDD have been reported [4,9,10,26], but the overall picture of binding is still not sufficiently consistent at the moment. Therefore, we suggest that the lack of a reliable SBDD means for SrtA leads to a lack of the developed SrtA inhibitors fulfilling the lead compounds’ requirements.

Due to the lack of a reliable means of SBDD-guided design of small-molecule lead compounds potentially developable into antivirulence drugs, a safer way is to start from a crucial part of natural substrates of SrtA—the “sorting sequence” LPxTG (Leu-Pro-Any-Thr-Gly). To date, two works have been published in this direction. The first one is the work by Wang et al. [2], where a short in silico screening led to a virtual hit-containing LPRDA (Leu-Pro-Arg-Asp-Ala) sequence, which was claimed to resemble the original LPxTG sorting sequence. The follow-up experimental check revealed the antivirulence activity of the PEG2000-LPRDA-NH2 molecule with an IC50 of 10.61 μM. The second recent study by Abujubara et al. [32] is well designed to develop the direction of small polypeptides resembling the sorting sequence of StrA substrates. In the work, several inhibitors were experimentally identified with IC50 values below 200 μM, with the most active molecule being LPRDSar with an IC50 of 18.9 μM, where Sar is sarcosine (N-methylglycine). Recently, it was experimentally confirmed that uncapped oligopeptide LPRDA results in reduced bacterial adhesion and biofilm formation in a dose-dependent manner [33], which suggests that even such a short sequence as LPRDA can present physiological effects and can be used either as is or as a starting point for further drug discovery.

Despite several short oligopeptides having been shown to reveal the experimental inhibition of SrtA, it is unlikely that oligopeptides will be used as such. Obviously, the subsequent optimization of the oligopeptides is necessary in two directions: the optimization of ligand efficiency and the optimization of ADMET (Absorption, Distribution, Metabolism, Elimination, and Toxicity) properties. Both the activity and ligand efficiency (LE) of the hit oligopeptides are suboptimal for drug purposes, since they have 40 to 50 heavy atoms and a molecular weight (MW) exceeding 500, yet show 2-digit micromolar activity, thus leading to an estimation of LE as less than 0.17 kcal/(mol·atom), appreciably lower compared to the reference value of LE of 0.3 kcal/(mol·atom) [34]. Another reason is that polypeptides are quickly degraded in living conditions, so that their bioavailability is generally low. Additionally, the above specific oligopeptides are heavily charged at physiological conditions, which contradicts the generally accepted assumption that the major driving force for drug–receptor interactions is the hydrophobic interaction [35,36]. Thus, these oligopeptides could be used as hit molecules in order to gradually replace inefficient and prone-to-degradation groups with more efficient and stable ones. The provided arguments are the rationale for the development of peptidomimetics [37,38], usually starting from an active but quickly degradable peptide in order to arrive at a drug with the desirable properties. This process should rely heavily on the receptor space model to make possible the use of the repertoire of structure-based drug discovery tools. However, creating a reliable structure source for SBDD, as it was pointed out earlier, is currently the main challenge for the rational development of SrtA inhibitors [39]. In particular, in the above-mentioned works of Wang et al. and Abajubara et al., the modeling part relied on the (as described earlier) problematic receptor structures, but it was enough to arrive at semi-quantitative conclusions.

To sum up, the intrinsic complexity of the system under study (SrtA-accessible conformational space) is further aggravated by the fact that simple, robust, and rational means of computer-aided drug discovery have not been established for SrtA to streamline further development. This is in striking contrast with SrtA as a target, which has long been validated as promising. We postulate that the pertinent model for SBDD should possess a partially contradictory set of requirements:It should reflect the different conformations of SrtA, reachable at physiological conditions;It should reflect conformations that are relevant for binding an inhibitor, assuming some of the conformations are not relevant despite being feasible at the relevant conditions;It should reflect conformations pertinent to different large-scale motions and rearrangements (e.g., transition “order–chaos” of the β6-β7 loop [13]) of the SrtA protein, not just the local flexibility caused by side-chain motion, regularly taken into account by simplified “flexible docking” protocols;It should be computationally cost-effective to allow for virtual screening and ideally lead to further optimization; at least, it should be significantly less demanding than the state-of-the-art molecular dynamics (MD) studies of microsecond time scales [12,13,14].

In this work, we aim to create a set of relevant conformations of SrtA for further use within the ensemble docking approach [40]. In the ensemble docking approach, a challenging and usually conformationally flexible receptor is represented with several distinct conformations during docking. Ensemble docking is more complex than docking to a single conformation of a receptor (the prevalent practice), but it is also several orders less resource-demanding than the complete MD simulation of the putative complexes. The intermediate between ensemble docking and the full MD study approach, the relaxed complex scheme, was reported to show promising results [41] for SrtA, but the scheme is still rather complex for massive screening and hit-to-lead optimizations of SrtA inhibitors. The ensemble docking approach was tested recently for modeling SrtA [39], showing mixed results. One of the useful conclusions of that work was that the geometries from the PDB structures of SrtA provide inferior power to classify molecules into actives and inactives compared to several MD-derived conformations. Another important observation is that none of the best performing conformations was able to properly classify all the true active molecules, leading to the suggestion that different conformations might be relevant to bind different molecules to SrtA.

We believe that the choice of conformations to use in ensemble docking is crucial. The conformations used in Ref. [39] were uniformly sampled from the MD trajectory and thus do not necessarily reflect the statistically weighted diversity of the conformations. An additional concern is the convergence of the trajectory used to produce conformations in terms of the large-scale motions pertinent to the protein under study.

In this work, we test a hypothesis that an ensemble docking approach with a specifically chosen set of conformations of SrtA receptors can be used as a pertinent SBDD tool to streamline in silico StrA inhibitor development, in particular to address the unmet need to support the rational hit-to-lead optimization of initial hit structures.

In what follows, we describe how we derive the representative set of SrtA conformations for further use in docking studies. Then, the selected conformations are used for subsequent ensemble docking. Both the conformation differences to known PDB structures of SrtA as well as the docking results obtained are analyzed and compared to the known experimental activities of several oligopeptides. Then, the developed docking approach is used to predict in silico the binding affinities for several proposed peptidomimetics, produced by the replacement of the lipophilic side-chain LP part of the LPRDA oligopeptide with the rather hydrophobic organic small-molecule residues developed in our laboratory [42]. Finally, the obtained results are discussed in a broader context, and the directions of future development are outlined.

## 2. Results

### 2.1. Representative Conformations of SrtA

Based on the analysis of the known facts about StrA and the efforts undertaken by researchers in the field, it was decided to generate conformations of SrtA that are the most relevant to the “sorting” sequence binding. Since the original sorting sequence LPxTG was not reported to bind or reveal inhibiting activity outside the long polypeptide chain, conformation sampling was conducted using the molecular dynamics (MD) study from our previous work, where LPRDA-SrtA complex geometries were extensively studied. Briefly, MD runs were started from different LPRDA-SrtA complex geometries obtained from AutoDock 4.2 and AutoDock Vina 1.1.2 docking. In that work, we showed that the binding of the LPRDA oligopeptide to SrtA significantly facilitates the conformational sampling of the receptor, including the loop β7-β8 and even the loop β6-β7, responsible for the significant change in the shape of the binding site and undergoing transitions “disorder–order” by accidentally forming a short alpha helix instead of the coiled structure of the loop [13]. Thus, LPRDA could be effectively considered as a binding site “plasticizer” upon binding. The effect of enhanced conformational plasticity was also noted earlier in the context of Hsp90-hAgo2 interactions [43]. We hypothesized that the main reason for the accelerated conformational sampling is that LPRDA forms numerous nearly equivalent energy hydrogen bonds with SrtA in quite different complex geometries, thus making different protein conformations more accessible at body temperature. It should be noted that the experimental NMR study confirmed that unbound LPRDA is represented with multiple conformations in water media [44]. Due to the above properties of LPRDA-SrtA complex MD trajectories, we hypothesized that different MD runs started from docking geometries after certain equilibration effectively sampled the same conformational space. For this reason, we decided to use all the available MD trajectories starting from different LPRDA-SrtA docking complex geometries as a source of the most abundant conformations in the statistical ensemble.

Eight different LPRDA-SrtA complexes obtained by both AutoDock 4.2 (three complexes) and AutoDock Vina 1.1.2 (five complexes) were described previously in detail [29] and were used to generate the SrtA conformation sampling in this work. Each docking complex geometry was used as a starting geometry in the MD studies, but with different random seeds, which resulted in quite different MD trajectories, even for the same initial complex as described elsewhere [29], thus generating additional statistics.

Each MD trajectory (10 ns in total) was represented with 100 equally spaced snapshots with intervals of 100 ps from each other, with no significant short-range autocorrelation assumed to be retained between the snapshots. Overall, 8 × 3 × 101 = 2424 snapshots were used for the cluster analysis.

Cluster analysis was performed via the GROMACS utility ‘cluster’ using the GROMOS clustering scheme, with a central structure having the smallest distances to all other members of a cluster. In total, 65 distinct clusters were found (with an RMSD cutoff equal to 1.3 Å). For this work, the six most representative (the most populated in the MD trajectories analyzed) clusters were selected, overall describing 59.5% of all the frames used for the analysis (Table 1).

Firstly, the extracted clusters reveal variability in the β7-β8 and especially in the β6-β7 loops (Figure 1), thus confirming that these elements are the most variable part in the binding site, which locates the His120-Cys184-Arg197 catalytic triad. Therefore, the clusters reasonably span the conformational space available for those loops.

Secondly, the geometries of clusters differ substantially from the initial StrA geometry (PDB:1T2W), which is in accord with a previous study [21]. The latter indicates that the PDB geometry may reflect just one of the available conformations for SrtA in relevant conditions.

Lastly, if the side-chain positions that form the binding site of SrtA are also taken into account, then, the differences between the clusters (and the initial structure as well) are even more pronounced. Figure 2 shows that the bindings sites of the cluster structures, with a significantly different volume, shape, and even electrostatic/hydrophobic nature, are formed. This corresponds to the results of previous studies, in which the adaptive nature of oligopeptide binding to SrtA via different binding modes was highlighted [12].

Our main hypothesis should be reiterated at this point. If a protein structure adopts multiple conformations with different binding site characteristics, the search for and the use of single geometry is not warranted, since multiple but feasible alternative binding options might be missed. With this perspective in mind and the observations outlined above, the obtained clustered geometries fit the purpose and the specific requirements put forward in the Introduction Section well.

### 2.2. Validation Using Different Oligopeptides

A validation of the extracted clusters and the approach itself using the known experimental data is an important step. The LPRDA oligopeptide was proposed and shown to possess antivirulence activity against *S. aureus* in the work by Wang et al. [2]. Despite the in silico modeling of this work being performed using the uncapped LPRDA sequence, the experimental check was performed using a capped version of the oligopeptide, which could affect the binding and hence our structure–activity interpretations. A more recent work by Abujubara et al. [32] intelligently expands on the initial findings by Wang et al. [2] by suggesting smart substitutions concerning the LPRDA oligopeptide in order to elucidate structure–activity relationships to uncover the underlying structural features important for the affine and hopefully selective binding of ligands to SrtA. The modified oligopeptide sequences used in this work were experimentally tested without additional sequence capping, so the results of this work are a more reliable basis to elucidate structure–activity relationships in terms of SrtA protein conformation. Therefore, nine structures (Figure 3) from Abujubara’s work presenting the known experimental activity (Table 2) were selected to test to what extent the variation in the experimentally observed activities of the selected structures could be explained using the ensemble docking approach proposed in this work.

By comparing the experimental activity and the obtained numerical results, it is possible to study the following hypotheses. Firstly, is there any difference in substituting positively charged Arg (R) in the middle of the sorting sequence with negatively charged Glu (E)? Secondly, to what extent is the addition of the more hydrophobic fragments at both ends of the sequence beneficial, as revealed by Abujubara et al. experimentally?

The protonation states of all the oligopeptides studied were obtained via OpenBabel v.3.0.0, where the pH = 7.

The analysis of the results (Table 3, Figure 4) of molecular docking to the set of SrtA conformations leads to the following observations. Firstly, it is seen that the representative clustered geometries studied result in lower (more beneficial) values of the predicted energy for oligopeptides **1**–**9** compared to the docking of the same oligopeptides to the most relevant SrtA models from PDB: 1T2W and 2KID. It shows that, at physiological conditions, there exist conformations of SrtA that are more amenable to the binding of ligands. Secondly, despite a certain variation, it is clearly seen that the model of Cluster #3 basically leads to the lowest attainable energies in our experiment, thus showing the most beneficial interactions with ligands. We argue that such preference is caused by the presence of the most compact and deep binding site among the other protein structures studied due to the fact that loop β6/β7 has a partially ordered form with one turn of α-helix compared to the more disordered (coiled) conformations present in the other protein structure studied (Figure 5). The described arrangement leads to the presence of the more pronounced hydrophobic regions, the latter being the major driver of ligand–receptor affinities.

Thirdly, it is clearly seen that the values of the ligand efficiency (LE, for short, the measure of efficiency of using heavy atoms in ligands) for the docked and scored complexes are basically lower than the threshold value of 0.3 kcal/(mol·atom) commonly adopted for estimating the binding efficiency of drug-like small molecules. The obtained LE values just reaches this threshold for ligands **1** and **2** bound to the Cluster #3 structure according to docking. On the one hand, the rather moderate values of LE neatly agree with the typical values related to the early stages of drug discovery, such as hit finding and hit-to-lead optimization. On the other hand, such values reflect the fundamental property of the relativity flat binding site (only part of the surface can take part in the interaction with a ligand), as well as the inherently lower efficiency of using heavy atoms pertinent to biopolymers compared to well-optimized small-molecule drugs and drug candidates.

One of the major advantages of ensemble docking is the possibility to reveal molecules more prone to binding to different conformations of the receptor. For that purpose, in this study, we calculated Boltzmann weights (e^−ΔG/RT^) for each of the combinations of ligands **1**–**9** and receptor geometry, using AutoDock4.2 scores as ΔG estimates, which were used in turn to calculate (Table 4) the probabilities of binding of each ligand to each of the SrtA geometries—*ρ_i_* from (1)—assuming they form a complete ensemble (that is, neglecting other accessible conformations, not explicitly studied in this paper). It is well seen that, in accord with the previous analysis, the studied oligopeptides **1**–**9** generally prefer to bind to Cluster #3. However, for structures **3**, **6**, **7,** and **9,** appreciable contributions of binding to other (not Cluster #3) SrtA geometries are observed. It is interesting to note that, for structures **3** and **6,** the more preferable binding is revealed for structures taken from PDB (with PDB:1T2W being the geometry used for docking and subsequent MD used for clustering). Note that precisely those structures contain Glu(E) instead of Arg(R) in the middle of the five membered amino acid oligopeptides studied. The experimental activities provided by Abujubara differ for Glu-containing structures **3** and **6**: whereas structure **3** shows moderate activity, structure **6** does not show it at the experimental conditions used. Additionally, it should be noted that the structures that are more prone to binding to different SrtA conformations studied are **3**, **6**, and to some extent **9**.
(1)$$\rho_{i}=\frac{e^{-\Delta G_{i}/RT}}{\sum_{j}e^{-\Delta G_{j}/RT}},$$
where *ρ_i_* is the weighted ensemble probability to encounter a complex of the probe ligand with the *i*-th SrtA geometry; Δ*G_i_* is the interaction energy (estimated by AutoDock4.2) of the probe ligand with the *i*-th SrtA geometry.
(2)$$ENE\left(BW\right)=\sum_{i}\Delta G_{i}\cdot \rho_{i}=\sum_{i}\Delta G_{i}\cdot \frac{e^{-\Delta G_{i}/RT}}{\sum_{j}e^{-\Delta G_{j}/RT}},$$
where *ENE*(*BW*) is the Boltzmann-weighted interaction free energy of the probe ligand among the ensemble of the SrtA geometries used.

Since an appreciable distribution of Boltzmann probabilities of binding between SrtA structures is observed, despite the clear dominance of binding to the Cluster #3 structure, the correspondence of the experimental activities and the predicted values should be determined using the weighted ensemble approach. The ensemble weighted (using Boltzmann weights) values of the scoring function of AutoDock4.2 (having the dimension of free energy) for the optimal binding positions, Ene(BW) (2), are presented in Table 4 for each studied ligand. A moderate correspondence to the experimental data activities is observed. The structures **2** and **5** are correctly predicted as the most active/affine. The structures **6** and **7** are correctly predicted as the least active. However, there are two significant cases of discrepancy in prediction. The reference structure **1** of unmodified LPRDA is incorrectly predicted to be active in contrast to experimental activity, almost exclusively due to exceptional binding to Cluster #3. The second case of discrepancy is structure **3** having its middle Arg(R) substituted with Glu(E) in the reference LPRDA oligopeptide ligand sequence, which is predicted as the worst overall binder to the SrtA structures studied. We suppose that the somewhat overrated predicted activity of the reference oligopeptide **1** (LPRDA) can be explained as a combination of two factors: a certain bias is introduced from taking the SrtA conformations from LPRDA-SrtA molecular dynamics and perhaps the presence of a conformation close to Cluster #3 in the actual ensemble of SrtA conformations is statistically somewhat overestimated. As for the Glu-containing structure **3** (LPETP), its underpredicted activity may be explained by the fact that the pattern of structural stabilization involved in adopting either Arg(R) or Glu(E) in the middle position of the reference LPRDA sequence is quite different, which is natural considering their opposite formal charges. It seems that conformations pertinent to well-binding polypeptide sequences with Glu(E) in the middle are not well presented in the six most abundant clusters chosen. As a solution for further development, the complexes including sequences with Glu(E) should be included in the MD simulation with the subsequent clusterization of SrtA geometries as well as, perhaps, using more than six cluster geometries to represent the SrtA conformation ensemble. It should be noted that the Glu(E)-containing structures **3** and **6** are predicted (Table 4) as not being dominated by the binding energy of the Cluster #3 SrtA geometry, as for the rest of the oligopeptide sequences. The latter suggest that the Cluster #3 structure is beneficial for binding sequences containing Arg(R) in the middle. Taking apart the described discrepancies, one can conclude that, for Arg(R)-containing oligopeptide sequences, the prediction is satisfactory, which is encouraging considering the complex nature of the highly flexible binding site target and the compact size of the cluster conformations used.

A separate notion worth noticing is the fact that the geometry of SrtA from Cluster #1 is represented in more than a fifth part of all snapshots of the analyzed MD trajectories. Still, the binding to Cluster #1 conformation is among the least favorable according to docking for most ligands. At a structural level, it is explained by the fact that the binding site in Cluster #1 is represented by a large and rather shallow cavity, which does not have a strong binding compared to deep and predominantly hydrophobic pockets. This observation seems to support the idea that only a fraction of the accessible SrtA conformations is relevant to affine binding. Even more interesting is that predictions made using only Cluster #1 (as the most abundant) qualitatively contradict the experimental observations. For instance, the structures **2** and **5**, most active in the experiment, are predicted as the least active (Table 3), whereas the experimentally least active structures **6** and **8** are predicted as active. Since the Cluster #1 conformation is the most represented in the MD simulation, it is possible to explain why the evenly taken snapshots of the MD trajectories, as used in the work in [21], do not form an optimal basis to build a set of reference conformations for ensemble docking for such a complex and adaptable structure as SrtA.

Additionally, it should be noted that the use of experimental models from PDB also does not provide a reliable basis for prediction. For both cases of PDB:1T2W and PDB:2KID as reference for docking, the discrepancy in the prediction of the active structures **2** and **5** and not active structures **6** and **8** is significant, as in the case of the Cluster #1 structure above. Moreover, for PDB:2KID, the most experimentally active structure **2** is predicted as being among the least active.

An analysis of the predicted geometry of ligand **2** (the best according to Abajubara research) with the SrtA structure of Cluster #3 reveals reasonable structural interaction patterns (Figure 6). Firstly, a good filling of the binding site with the ligand, with well-pronounced intermolecular contacts, is observed. Secondly, the electrostatic complementarity is also very good. On the one hand, the Arg(R) of ligand LPRDSar (**2**) forms a complete (two hydrogen bonds) salt bridge with Glu105 and a single hydrogen-bond contact with the Glu105 of SrtA. Additional hydrogen bonds are formed with the carbonyl oxygens of the backbone residues of Asn114 and Gln172. Thus, the docked structure suggests that the Arg (R) coordination of the ligand imitates the Ca^2+^ binding with the residues of SrtA specifically dedicated for it [14]. On the other hand, the side-chain of Arg197 of SrtA is well coordinated with the carboxylic group of Asp (D) and by the backbone carbonyl oxygens of Pro (P) and Asp (D) of the ligand. The interaction of such type with Arg197 was postulated by us earlier [22] as important on the basis of the comparison of the experimental data available and the results of the simulations. Consequently, we put forward a hypothesis that not only the Cluster #3 geometry forms a pronounced pocket in the binding site of SrtA, facilitating binding, but also makes it possible to form multiple charged hydrogen bonds with the catalytic residue Arg197; the importance of the interaction of a ligand with the latter was shown earlier. Overall, the analysis of the structures of the complexes shows the validity of the structures of clusters (representative geometries of SrtA) for further use in predicting the binding affinity (activity) for other test molecules of ligands.

Thus, the initial approbation of the entire approach showed its relevance to the outlined tasks of structure-based design modeling. Despite the revealed dominance of the interactions from Cluster #3, for several ligands, appreciable corrections from interaction with other SrtA conformations are observed. The approach may thus be considered as validated to predict the affinity of other ligands.

### 2.3. Modeling KUD Peptidomimetics

#### 2.3.1. Ligands

In conditions when molecular modeling still does not point to a definitive direction of design and/or a modification of the existing hits, hit modification is conducted guided by medicinal chemistry considerations and synthetical accessibility of analogs. Thus, it has been suggested that replacing the relatively nonpolar part, LP, in the LPRDA ligand with one of the low-molecular frameworks from the KUD series of compounds is promising. This substitution has several goals. Firstly, the KUD family of structures generally has rather rigid frameworks, which might be useful for both increasing the affinity to SrtA and to enhance selectivity to off-targets. Secondly, the presence of Pro in an amino acid sequence is known to lead to a characteristic turn in a secondary structure due to intrinsic chirality and the relative rigidity of the ring fragment. This is one of the reasons behind the choice of KUD series structures to replace LP fragments: they are generally able to form a similar turn and most of them are chiral. Thirdly, one of the main reasons to use peptido-mimetics instead of the oligopeptide sequences is the enhanced bioavailability of the former due to the increased times of biodegradation. Native oligopeptide sequences are prone to quick metabolism in living systems, which is why the appreciable modification of oligopeptides can lead to diminished biodegradation.

In this work, six KUD structures were investigated to replace the LP part of the reference LPRDA sequence. Each of the structures is represented with two distinct enantiomers (Figure 7).

Since the amino acids Leu (L) and Pro (P) possess chirality, which affects the secondary and tertiary structures of polypeptides, it is natural to assume that, in a replacement scheme **LP**-RDA -> **KUD**-RDA, certain enantiomers can have a significant preference in binding to the generally chiral SrtA binding site. One of the goals of the current work was to assess to which extent the choice of KUD enantiomer in **KUD**-RDA ligands affects the predicted affinity. To this end, each investigated structure was explicitly represented with two enantiomers, and the stereochemical configuration of RDA amino acids was in all cases fixed to L-isomers, the most abundant in nature.

#### 2.3.2. Results of Docking

The energies of the best docking poses of the studied KUD-RDA ligands with each of the presented geometries of SrtA are presented in Table 5 and Figure 8. It can be seen that the key findings revealed in the stage of the validation of the proposed procedure for ensemble docking using ligands **1**–**9** from the Abajubara work are quite similar to the observed results for ligands **10**–**21** from the KUD-RDA series.

In particular, the distribution of the obtained energies shows a similar pattern. Firstly, generally, the most efficient interactions are formed with SrtA in the Cluster #3 geometry. For the same SrtA geometry, the ligand efficiencies (LEs) that are the closest to the commonly acceptable threshold of 0.3 kcal/(mol·atom) are attained (Table 5, LE columns colored as proximity to the threshold value). Secondly, generally, the least beneficial interactions are formed with the Cluster #1 and Cluster #6 structures of SrtA, which are the most and least represented in the MD trajectory, respectively.

An analysis of the distribution of the probability to bind each particular SrtA structure suggests that, despite the dominance of Cluster #3 binding, an appreciable binding can occur also with other SrtA geometries (Table 6). Similar to the results of the validation set **1**–**9,** the contributions from the PDB structures of SrtA (1T2W and 2KID) rarely prevail (a notable expectation is structure **16**); however, they still make appreciable Boltzmann contributions to the ensemble.

For structures **14**, **17**, **18**, **20,** and **21** (the darkest green highlights in Table 6), the predicted Boltzmann-weighted binding energy values are comparable to or better than predicted for the reference structure LPRDSar (**2**). Therefore, these structures should be prioritized based on the results of modeling. Moreover, it should be noted that these structures are represented in each case with a single enantiomer of KUD-RDA. This confirms that the binding site is susceptible to the stereochemistry of ligands, provided they are not too small. It additionally validates the idea of the design used to replace LP with KUD fragments, since in both cases the chiral influence is appreciable.

The structural analysis of the most favorable predicted complex of **14** with SrtA in the Cluster #3 geometry shows (Figure 9) that the fragment KUD225-5R of the studied peptidomimetic fits almost perfectly the hydrophobic pocket formed by the rigid β-strand of the core β-barrel structure of SrtA and the folded geometry of the generally flexible β6/β7 loop. Other crucial elements described earlier in the analysis of Cluster #3 binding are basically conserved in the complex, but in somewhat modified forms. Thus, the carboxylic group of the side-chain of Asp (D) as well as the terminal carboxylate of Ala (A) of RDA sequence essentially “chelate”, with their interactions, the catalytically important Arg197, according to predictions. The Arg (R) of the RDA sequence, in turn, coordinates well with Glu108, Glu105, as well as with Asn114 and Gln172 in a practically identical structural position compared to the predicted geometry of the experimentally best structure **2** of the Abujubara set.

Despite the current study revealing a good correspondence of the crucial binding factors for the peptidomimetics under question, in the predicted complexes, there is still room for additional rational optimization. In particular, a part of the site surface formed by the β3-strand, the side-chain of Arg197, leading to the catalytic Cys184, remains unused for direct interactions. There is potential to improve both the affinity and selectivity of the studied molecules by forming direct interactions with the described area. Another optimization possibility is to form covalent rings in places where the predicted ligand geometry forms intermolecular hydrogen bonds in order to fix the active conformation and thus reduce the penalty for entropy loss caused by the great number of freely rotatable bonds characteristic of oligopeptide fragments.

Thus, a combination of the structural and energetic analyses of the docking results of the studied KUD-RDA peptidomimetics to the ensemble of SrtA geometries shows that the peptidomimetics reasonably reproduce the crucial patterns of interactions revealed in the analysis of the binding of the Abujubara structures **1**–**9** as well as introduce certain additional interaction patterns. On the other hand, it was shown that the ensemble docking approach for modeling SrtA interactions at the structural level (SBDD) with potential inhibitors has good applicability and has prospects for further rational structure-based drug discovery in the field.

## 3. Discussion

The design of our work and the obtained results are in the spirit of the tendency to tackle traditionally “tough” targets [40,45,46] using the structure-based drug discovery arsenal of approaches and tools, which is on the rise in contemporary science. In particular, targets that are hardly representable with a single relevant conformation are being studied with ensemble approaches. Recently, Stachowski et al. [47] studied the heat-shock protein 90 (Hsp90α) as a promising anticancer target. It was shown that the experimentally available conformations of the protein (complexed with chemically different ligands) could be clustered into three main cluster representative structures differing mainly in the conformation of the part of the flexible binding site—the “lid”. It was also shown that the resulting conformations possess different “hotspot” patterns and can therefore bind chemically distinguishable ligands. This corresponds well to SrtA targeting. For other challenging and important drug targets, such as kinases, the ensemble generation with subsequent docking has been successfully used. For example, for the c-Met target, the binding site’s unusual plasticity was explained using the ensemble approach and a potential c-Met inhibitor exploiting previously unseen binding modes was also proposed [48,49]. We believe the ensemble approach is crucial to consider when modeling SrtA interactions, since to date chemically and structurally different ligands have been shown to inhibit SrtA [4,9] that are unlikely to bind in the same binding site configuration.

At the same time, it was shown in the work of Gao et al. [39] that neither the experimentally available nor MD-generated single conformation of SrtA are able to discern actives from decoys in a docking study, with several actives preferring certain SrtA conformations. The authors also found that the more compact SrtA conformations are in terms of the distance from the catalytic Csy184 to the flexible β6/β7 loop, the more enrichment is obtained for different ligands. This is in accord with our results, where the Cluster #3 geometry (having the most compact arrangement of the β6/β7 loop relative to other site elements, in particular, Cys184 position) creates the most favorable interactions according to docking, since that geometry better resembles “good” small-molecule binding sites, in contrast to other conformations, which are more “open” and possess, therefore, a binding site that is more similar to the large and flat binding sites encountered in protein–protein interactions (PPIs) [30,31].

In this work, a proof-of-concept study is presented aimed at assessing the ensemble method, which does not add significant additional computational burden compared to a single-receptor conformation docking. Evidently, an increase in the number of the clustered conformations or even a use of non-clustered snapshots from a relatively long MD simulation, as suggested in the work of Evangelista et al. [50], will result in a better coverage of SrtA conformation, potentially relevant to describe ligand-SrtA interactions. However, it is hard to decide beforehand how many conformations suffice when dealing with a target such as SrtA, since they significantly change the binding surface (hence, the interaction patterns) and are not as densely covered by reliable experiment inhibitors as was the case for the targets from the study of Evangelista. An additional point for the further development of the approach is to include, in the initial snapshot set, the conformations resulting from the MD simulation of complexes of LP**E**XX (of which structure 3, LPETP, was experimentally shown as being active by Abujubara [32]) with SrtA. It seems that StrA conformations, which could describe apparently different binding patterns (compared to LP**R**DA and analogs) well [11], are missing in the currently chosen cluster representatives. However, it is also possible that certain SrtA conformations (beyond #6) could represent such interactions well. This constitutes the direction for further development.

An additional point for development is the estimation of the probability of the SrtA conformation in the ensemble of conformations reachable at body temperature. The current study assumed for simplicity that each of the two PDB structures and obtained six cluster geometry had equal a priori probabilities to encounter them, which is evidently not the case in general.

Overall, the presented approach corresponds well to the known scientific background and is by design not resource-demanding to enable its direct use in applied drug discovery projects either in its current state or after certain enhancements, likely along the lines outlined above.

## 4. Materials and Methods

### 4.1. Molecular Dynamics of LPRDA-SrtA Complexes

To sample SrtA conformations for subsequent clustering, a molecular dynamics (MD) simulation of LPRDA-SrtA complexes was performed. The details of the choice of complexes and MD simulation are described elsewhere [29], where it was revealed that simulating the LPRDA-SrtA complex effectively leads to an accelerated sampling of SrtA conformations compared to the simulation of the apo form. The latter was attributed to the property of LPRDA to act as a “plasticizer” in this system. Eight different LPRDA-SrtA complex geometries obtained by docking (three with AutoDock 4.2 and five with AutoDock Vina 1.1.2) were used as initial structures for the MD simulation. Additionally, each initial geometry was used with 3 different random seeds, which resulted in appreciably different dynamics, hence creating an additional sampling of the SrtA geometry. The significantly different MD trajectories agree well with the partially chaotic behavior of the system caused by the intrinsically disordered regions of the loops β7-β8 and especially β6-β7 [13]. Each MD simulation lasted 10 ns after equilibration. In the current work, the MD trajectories obtained in the previous work [29] and described above were used as input for further clusterization of the conformational space of SrtA.

### 4.2. SrtA Conformation Clusterization

The snapshots for subsequent clustering were extracted from each MD trajectory as evenly spaced with 100 ps between them. Overall, 8 (initial complexes) × 3 (repetitions with different random seeds) × 101 (images in each MD trajectory) = 2424 snapshots were used for the cluster analysis. Cluster analysis was performed via GROMACS utility ‘cluster’ using the GROMOS clustering scheme with a central structure having the smallest distances to all other members of a cluster. For clustering, only the coordinates of the backbone Cα-atoms were used, which is usual practice. Overall, 65 distinct clusters were found (with RMSD cutoff equal to 1.3 Å). It is important to note that in the obtained clusters are represented with an actual structure from a certain MD snapshot with all distances and angles properly defined, not with mathematically mean coordinates (which may result in significantly unnatural local structures). Out of the 65 overall clusters obtained, only 6 of the most represented (covering 59.5% of all frames) were retained (Table 1) as representative ensemble geometries for subsequent docking and analysis.

### 4.3. Ligand Preparation

All of the used ligands were sketched with PyMol 2.5 (for regular protein sequences) and Avogadro 1.93 [51] for small-molecule (KUD) and non-standard amino acids. Space (3D) geometries of all molecules were generated from scratch using SMILES representation (with correct stereocenter information) as input and OpenBabel v3.0.0 as the means to produce spatial geometry (option --gen3D) [52]. The protonation states of all ligands were set to correspond to pH = 7 using the -p option of OpenBabel. MGLTools v1.5.6 [53] with the standard settings used to obtain the PDBQT-files necessary for subsequent AutoDock modeling.

### 4.4. Docking

The AutoDock 4.2 scoring function and docking was used as implemented in the AutoDock Vina 1.2.3 software [54], which enables parallel runs. The grid box was of size 18.75 × 18.00 × 23.25 (X × Y × Z) Å and centered in −37.0, −18.2, 4.3, which was chosen visually to include the binding site in all studied conformations of SrtA simultaneously (Appendix A). The coordinates of all eight SrtA conformations used in the work aligned to the same reference frame (using *cealign* utility of PyMol) and compatible with the grid box described above are provided in the Appendix A. The potential grids were generated using the *autogrid4* utility with the grid parameters specified above. The AutoDock Vina 1.2.3 software with parameter --*scoring = ad4* (to use AutoDock4 scoring function) and increased value of --*exhaustiveness = 64* was used to conduct docking.

### 4.5. Ensemble Analysis

AutoDock 4.2 produces the scores in free energy units enabling the direct use of statistical mechanics estimation of probabilities of complexes in the ensemble, using (1) and (2). The best energy docked complexes were used to produce ensemble statistics. Such a use assumes that only the explicitly considered SrtA geometries form the ensemble, whereas the other geometries result in significantly less beneficial complexes, leading to their negligible contribution to the ensemble.

## 5. Conclusions

In the conducted work, we tested the hypothesis of the relevance of the ensemble docking approach to obtain a reasonably reliable means of structure-based modeling (SBDD) of the interactions of SrtA with small molecules—potential inhibitors. It should be noted that such means have been lacking to date. As a result, we put forward and tested the algorithm of SrtA conformation selection, which makes it possible to meet the requirements for the SrtA structure, taking into account the pitfalls revealed previously in our works and by other researchers in the field.

It was shown that the ensemble docking approach leads to a qualitatively better description of the experimentally observed relationships in the binding of the series of LPRDA analogs (taken from Abujubara [32]). Additionally, one of the ensemble conformations of StrA (Cluster #3), which is present only in 7% of the MD snapshots used for the analysis, possesses a binding site with a partially ordered state of the generally flexible β6/β7 loop, with qualities approaching the characteristics of the “good” binding sites for small molecules due to the formation of a relatively deep and compact cavity compared to the initial experimental structure PDB:1T2W. Unsurprisingly, for many of the studied ligands, the use of precisely this conformation for docking leads to the most beneficial predicted interaction energy. However, there are cases of ligands where other conformations contribute significantly or even dominantly. Additionally, the Boltzmann-weighted contributions of the other ensemble SrtA conformations are appreciable and valuable for many of the ligands studied.

Using the validated ensemble approach, we obtained predictions for the series of KUD-RDA peptidomimetics in which the LP part of LPRDA oligopeptide was substituted with relatively rigid, chiral, and partially lipophilic fragments of KUD series compounds. The most prospective structures were selected for further experimental evaluation.

It was shown that, despite the proposed approach not being able to completely address all the difficulties associated with modeling of such a complex target as SrtA, it does form a more reliable basis for structure-based modeling compared to the approaches used previously. The latter were obtained by using the available PDB structures or via the ensemble approach, in which random MD snapshots were taken to build an ensemble of conformations. This result, in turn, makes it possible to use the proposed ensemble approach for the modeling of the key interactions of the developed small molecules and/or peptidomimetics aimed at designing new antivirulence drugs to combat hospital infections, in particular, *Staphylococcus aureus*, including MRSA and other resistant strains.

## Figures and Tables

**Figure 1 ijms-25-11279-f001:**
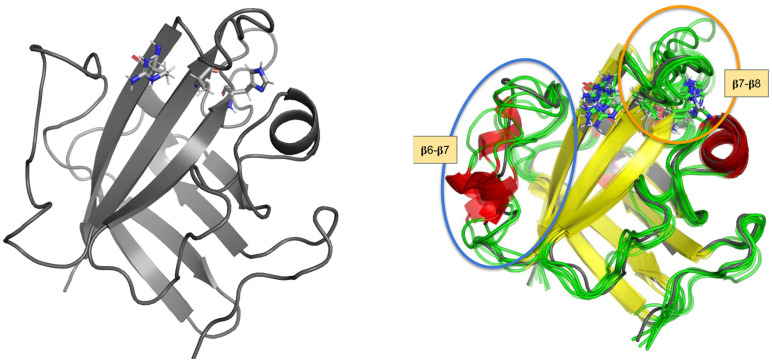
Structures of SrtA: (**left**)—initial structure from PDB:1T2W, (**right**)—superposition of the initial structure and 6 clusters used in this work, with a secondary structure color scheme. The key catalytic residues His120, Cys(Ala)184, and Arg197 are shown. The regions of the flexible loops—β6-β7 and β7-β8—are highlighted with ovals.

**Figure 2 ijms-25-11279-f002:**
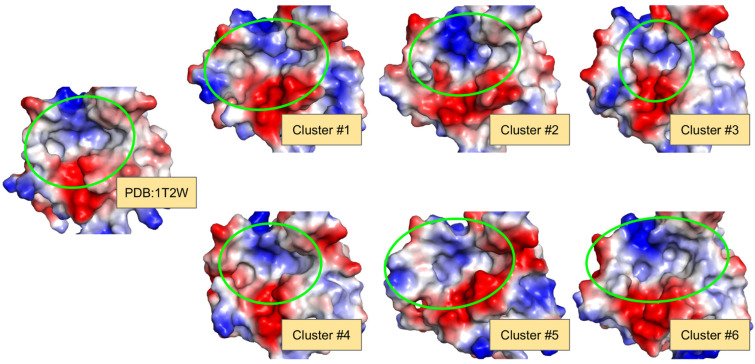
Surface representation of the initial (PDB:1T2W) and the obtained cluster structures reveals remarkable differences, resulting in binding sites with appreciably different shapes and electrostatic/hydrophobic nature. The surface with electrostatic potential was colored in PyMol 2.5, with blue being positive potential and red negative potential. The approximate site location and shape are highlighted with green circles.

**Figure 3 ijms-25-11279-f003:**
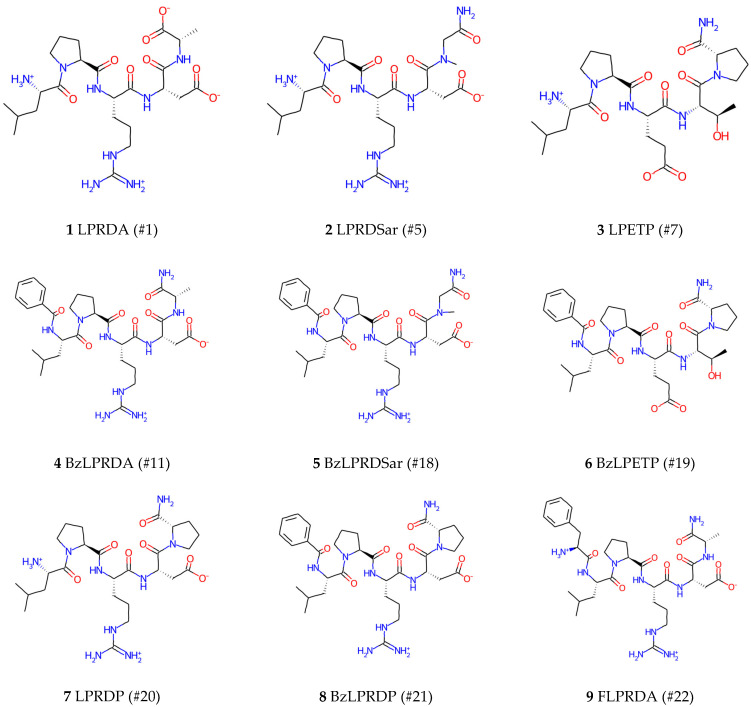
The molecules taken from Abujubara et al. [16] (with the names and numbers of structures provided for reference) for the protocol validation.

**Figure 4 ijms-25-11279-f004:**
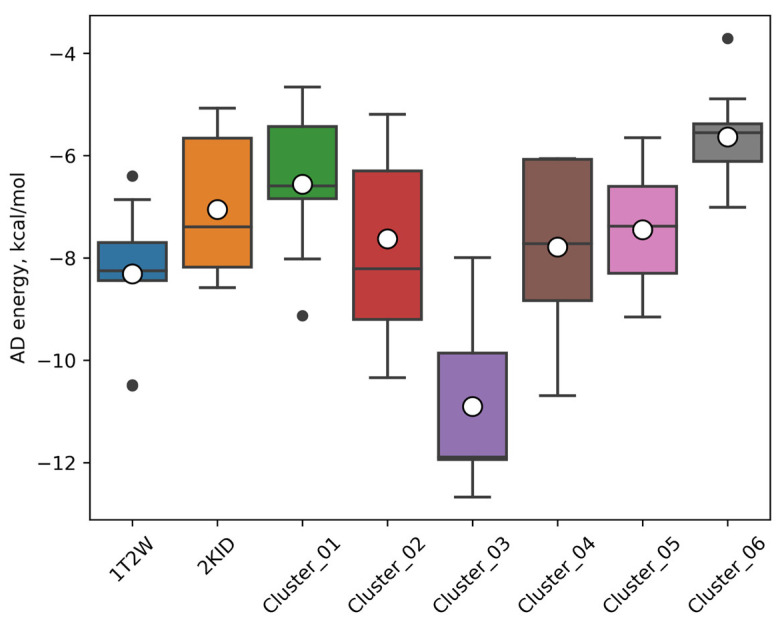
Distribution of the predicted AutoDock energies for the nine structures from the Abujubara set on the different structures of the SrtA protein. Horizontal lines—quartiles, empty circles—the mean values, and black circles—the outliers.

**Figure 5 ijms-25-11279-f005:**
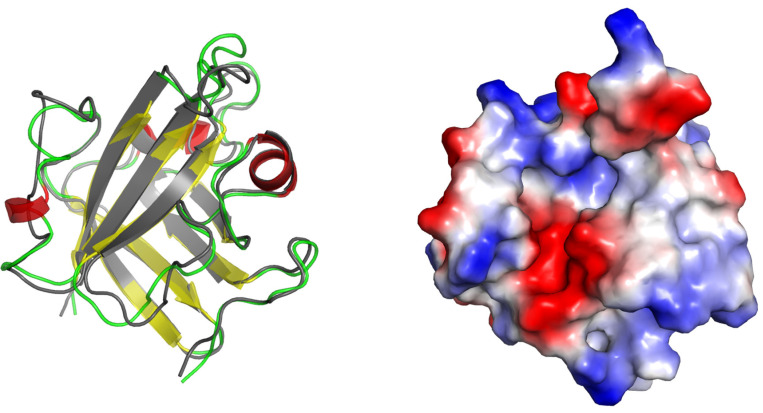
The structure of Cluster #3, revealing the most beneficial interactions with the ligands studied. (**Left**)—comparison of the secondary structure (backbone) of PDB:1T2W (gray) and Cluster #3 (colored according to the secondary structure) from our study. (**Right**)—the surface of Cluster #3 colored according to the electrostatic potential (using PyMol, with blue being positive and red negative values).

**Figure 6 ijms-25-11279-f006:**
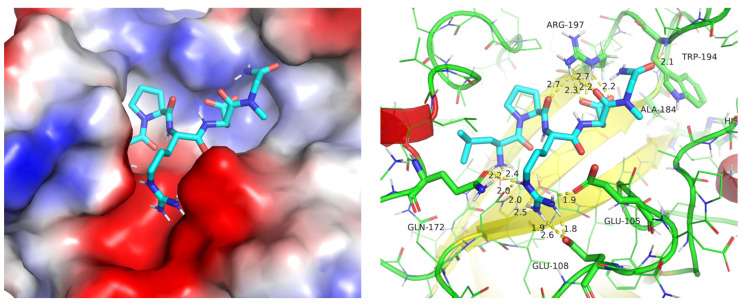
The predicted structure of complex **2**-SrtA with the geometry of SrtA taken from Cluster #3: (**left**)—the surface representation of the protein colored in electrostatic potential (red being negative and blue—positive values, obtained by PyMol), (**right**)—the cartoon and sticks representation of the protein colored in colors of the secondary structure (yellow—β-sheets, red—helices and green—coil).

**Figure 7 ijms-25-11279-f007:**
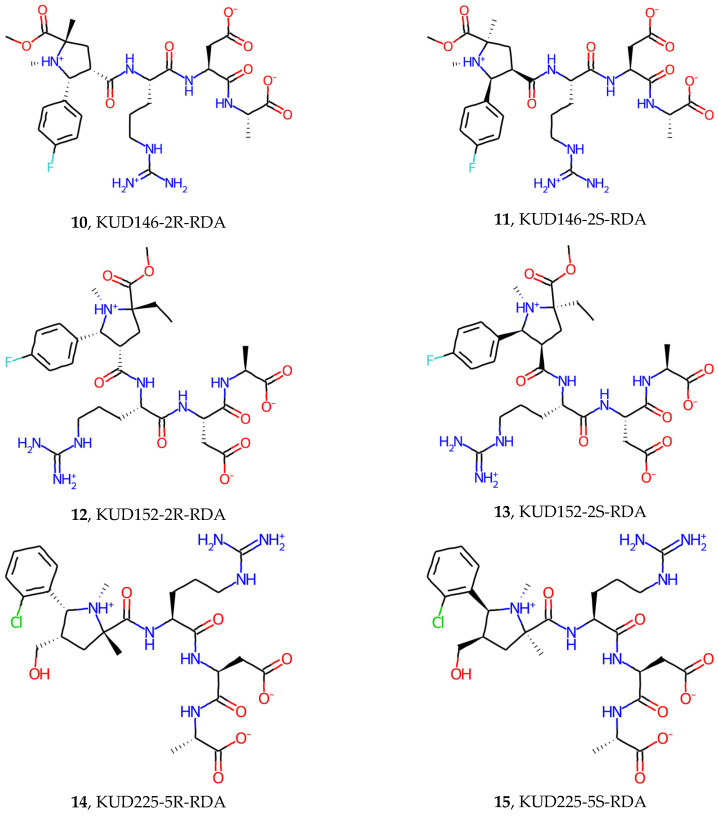
The studied KUD-RDA molecules with the configuration of each key stereocenter explicitly indicated.

**Figure 8 ijms-25-11279-f008:**
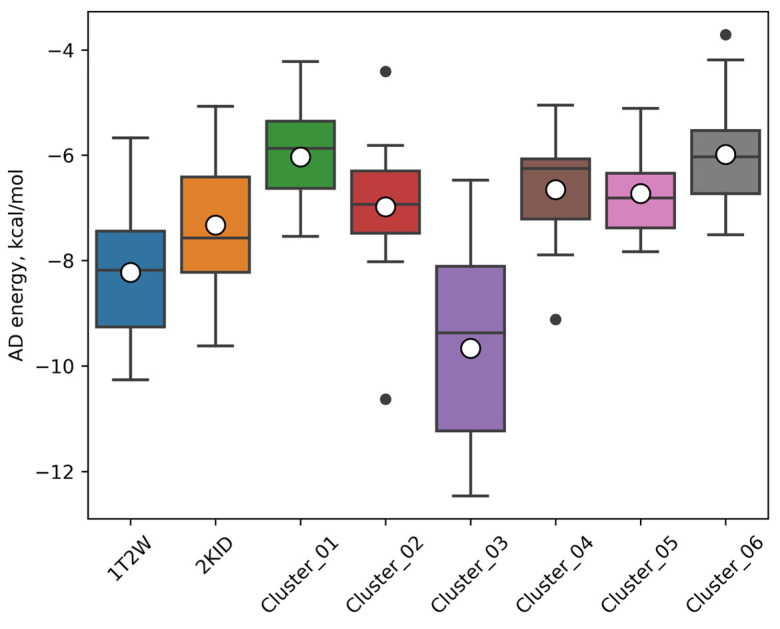
Distribution of the predicted AutoDock energies for the **10**–**21** KUD-RDA structures on the different structures of the SrtA protein. Horizontal lines—quartiles, empty circles—the mean values, and black circles—the outliers.

**Figure 9 ijms-25-11279-f009:**
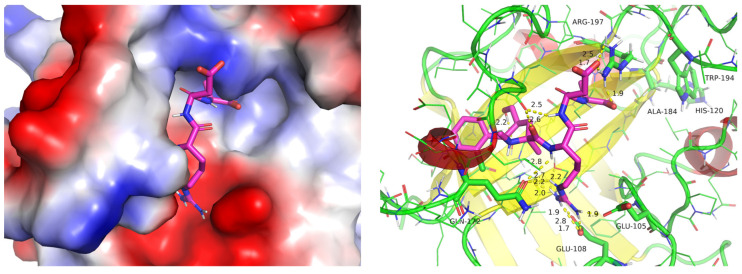
The predicted structure of complex **14**-SrtA (Cluster #3): (**left**)—the surface representation of the protein colored in electrostatic potential (red being negative and blue—positive values, obtained by PyMol), (**right**)—the cartoon and sticks representation of the protein colored in colors of the secondary structure (yellow—β-sheets, red—helices and green—coil).

**Table 1 ijms-25-11279-t001:** Number of MD frames assigned to each of the first six selected cluster centers. The total number of frames used for analysis was 2424.

Cluster #	Number of Frames	Percent of Frames
1	518	21.4
2	405	16.7
3	161	6.6
4	154	6.4
5	126	5.2
6	77	3.2
Sum	1441	59.45

Cluster #—is the Cluster number.

**Table 2 ijms-25-11279-t002:** Experimental activities of the selected nine structures from the work of Abujubara et al. [32]. Percent of inhibition for the FRET method is for the ligand concentration of 200 µM.

#	FRET Inhibition, %	IC50, µM	−RT ln IC50, kcal/mol
1	29 (3)	-	-
2	69 (3)	19	−6.52
3	76 (8)	136	−5.34
4	37 (3)	-	-
5	108 (7)	57	−5.86
6	15 (4)	-	-
7	32 (9)	-	-
8	31 (8)	-	-
9	70 (6)	185	−5.16

**Table 3 ijms-25-11279-t003:** Computed AutoDock 4.2 scores (E, kcal/mol) for the best docking positions of the oligopeptides studied in this work.

	1T2W		2KID		Cluster_01		Cluster_02		Cluster_03		Cluster_04		Cluster_05		Cluster_06		
num	E	LE	E	LE	E	LE	E	LE	E	LE	E	LE	E	LE	E	LE	NH
1	−6.86	0.17	−5.07	0.13	−5.27	0.13	−6.30	0.16	−11.94	0.30	−6.07	0.15	−7.38	0.18	−3.71	0.09	40
2	−8.41	0.21	−5.60	0.14	−4.66	0.12	−9.31	0.23	−11.92	0.30	−6.07	0.15	−8.89	0.22	−5.49	0.14	40
3	−6.40	0.16	−8.18	0.21	−6.49	0.17	−6.51	0.17	−7.99	0.20	−6.06	0.16	−6.60	0.17	−5.55	0.14	39
4	−7.76	0.16	−7.39	0.15	−6.59	0.14	−9.20	0.19	−11.89	0.25	−6.71	0.14	−5.65	0.12	−5.38	0.11	48
5	−8.25	0.17	−7.44	0.15	−6.60	0.14	−8.41	0.18	−12.28	0.26	−9.76	0.20	−6.42	0.13	−6.11	0.13	48
6	−10.50	0.22	−8.21	0.17	−9.13	0.19	−10.34	0.22	−9.68	0.21	−8.19	0.17	−7.10	0.15	−7.00	0.15	47
7	−8.44	0.20	−5.66	0.13	−5.43	0.13	−5.19	0.12	−9.86	0.23	−8.83	0.21	−7.56	0.18	−5.60	0.13	42
8	−10.48	0.21	−8.58	0.17	−8.02	0.16	−8.21	0.16	−12.67	0.25	−10.69	0.21	−8.30	0.17	−7.01	0.14	50
9	−7.70	0.15	−7.34	0.14	−6.84	0.13	−5.19	0.10	−9.90	0.19	−7.72	0.15	−9.15	0.18	−4.89	0.10	51

NH—the number of heavy (non-hydrogen) atoms of each ligand of **1**–**9** used to calculate LE using LE = −E(ΔG)/NH. The green color intensity in LE values corresponds to the closeness of the values to the standard threshold value of LE being 0.3 kcal/(mol·atom).

**Table 4 ijms-25-11279-t004:** Estimated probabilities of binding the oligopeptides of the studied set to each of the SrtA structures used in the study (including two reference PDB structures), calculated using Boltzmann weights separately for each ligand.

#	1T2W	2KID	Cl_01	Cl_02	Cl_03	Cl_04	Cl_05	Cl_06	Ene(BW) *
1	0.000	0.000	0.000	0.000	0.999	0.000	0.001	0.000	−11.94
2	0.003	0.000	0.000	0.013	0.978	0.000	0.006	0.000	−11.86
3	0.026	0.496	0.030	0.031	0.361	0.014	0.036	0.006	−7.86
4	0.001	0.001	0.000	0.011	0.987	0.000	0.000	0.000	−11.85
5	0.001	0.000	0.000	0.002	0.982	0.015	0.000	0.000	−12.23
6	0.460	0.010	0.047	0.352	0.117	0.010	0.002	0.001	−10.23
7	0.073	0.001	0.000	0.000	0.769	0.140	0.017	0.001	−9.56
8	0.024	0.001	0.000	0.001	0.938	0.035	0.001	0.000	−12.54
9	0.019	0.010	0.004	0.000	0.737	0.019	0.210	0.000	−9.62

* Ene(BW)—the Bolzmann-weighted mean energy of each compound interaction with the ensemble of geometries of SrtA used, with the green highlights corresponding to the most favorable energies and the red ones to the least favorable. The intensity of the probability values (columns except Ene(BW)) color corresponds to the probability values to enhance visual inspection.

**Table 5 ijms-25-11279-t005:** Docking results for KUD-RDA peptidomimetics.

	1T2W		2KID		Cluster_01		Cluster_02		Cluster_03		Cluster_04		Cluster_05		Cluster_06		
num	E	LE	E	LE	E	LE	E	LE	E	LE	E	LE	E	LE	E	LE	NH
1	−6.86	0.17	−5.07	0.13	−5.27	0.13	−6.30	0.16	−11.94	0.30	−6.07	0.15	−7.38	0.18	−3.71	0.09	40
10	−9.26	0.21	−8.22	0.18	−7.54	0.17	−7.48	0.17	−9.37	0.21	−7.89	0.18	−5.11	0.11	−5.27	0.12	45
11	−6.98	0.16	−6.05	0.13	−5.87	0.13	−5.95	0.13	−8.11	0.18	−5.41	0.12	−6.05	0.13	−6.78	0.15	45
12	−8.18	0.18	−9.05	0.20	−6.83	0.15	−7.14	0.16	−9.31	0.20	−7.07	0.15	−7.57	0.16	−6.40	0.14	46
13	−5.67	0.12	−5.86	0.13	−7.00	0.15	−4.41	0.10	−6.47	0.14	−5.81	0.13	−5.34	0.12	−4.19	0.09	46
14	−10.26	0.24	−7.01	0.16	−5.35	0.12	−6.37	0.15	−12.46	0.29	−6.70	0.16	−7.76	0.18	−7.24	0.17	43
15	−9.02	0.21	−6.41	0.15	−5.56	0.13	−6.60	0.15	−9.79	0.23	−9.12	0.21	−7.27	0.17	−6.73	0.16	43
16	−8.52	0.19	−6.45	0.15	−4.22	0.10	−7.13	0.16	−7.73	0.18	−5.05	0.11	−7.83	0.18	−6.03	0.14	44
17	−7.44	0.17	−7.57	0.17	−5.19	0.12	−10.63	0.24	−9.32	0.21	−6.25	0.14	−6.44	0.15	−5.53	0.13	44
18	−8.06	0.18	−7.86	0.17	−6.63	0.15	−6.93	0.15	−11.23	0.25	−7.21	0.16	−6.81	0.15	−6.03	0.13	45
19	−7.62	0.17	−7.69	0.17	−6.62	0.15	−5.81	0.13	−7.84	0.17	−6.14	0.14	−6.34	0.14	−5.84	0.13	45
20	−9.37	0.22	−9.62	0.22	−5.75	0.13	−8.02	0.19	−11.38	0.26	−7.60	0.18	−6.57	0.15	−6.57	0.15	43
21	−9.67	0.22	−8.37	0.19	−6.58	0.15	−7.96	0.19	−10.71	0.25	−6.21	0.14	−7.01	0.16	−7.51	0.17	43

NH—the number of heavy (non hydrogen) atoms of each ligand of **1**, **10**–**21** used to calculate LE using LE = −E(ΔG)/NH. The green color intensity in LE values corresponds to the closeness of the values to the standard threshold value of LE being 0.3 kcal/(mol·atom).

**Table 6 ijms-25-11279-t006:** Estimated probabilities of binding the KUD-RDA ligands to each of the SrtA structures used in the study (including two reference PDB structures), calculated using Boltzmann weights separately for each ligand.

#	1T2W	2KID	Cl_01	Cl_02	Cl_03	Cl_04	Cl_05	Cl_06	Ene(BW) *
2	0.003	0.000	0.000	0.013	0.978	0.000	0.006	0.000	−11.86
10	0.388	0.069	0.022	0.020	0.462	0.040	0.000	0.000	−9.11
11	0.109	0.023	0.017	0.020	0.722	0.008	0.023	0.078	−7.69
12	0.079	0.335	0.008	0.014	0.519	0.012	0.029	0.004	−8.99
13	0.058	0.079	0.527	0.007	0.219	0.073	0.033	0.005	−6.55
14	0.025	0.000	0.000	0.000	0.974	0.000	0.000	0.000	−12.40
15	0.169	0.002	0.001	0.003	0.614	0.199	0.009	0.004	−9.48
16	0.578	0.018	0.000	0.057	0.153	0.002	0.182	0.009	−8.12
17	0.004	0.005	0.000	0.889	0.099	0.001	0.001	0.000	−10.46
18	0.005	0.004	0.000	0.001	0.988	0.001	0.001	0.000	−11.19
19	0.248	0.276	0.047	0.012	0.354	0.021	0.029	0.013	−7.56
20	0.032	0.048	0.000	0.003	0.914	0.002	0.000	0.000	−11.21
21	0.145	0.017	0.001	0.008	0.823	0.000	0.002	0.004	−10.47

* Ene(BW)—the Bolzmann-weighted mean energy of each compound interaction with the ensemble of geometries of SrtA used, with the green highlights corresponding to the most favorable energies and the red ones to the least favorable. The intensity of the probability values (columns except Ene(BW)) color corresponds to the probability values to enhance visual inspection.

## Data Availability

Data are contained within the article or Appendix A.

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
