# Peer review of "Ensemble Docking as a Tool for the Rational Design of Peptidomimetic Staphylococcus aureus Sortase A Inhibitors"

_ijms, 2024, doi:10.3390/ijms252011279_

Round 1
Reviewer 1 Report
Comments and Suggestions for Authors
ijms-3257116, Ensemble docking as a tool for rational design of peptidomimetic Staphylococcus Aureus sortase A inhibitors
The article has a good value, but it could be improved by aplling some corrections to its editing and its content.
Explain shortly what LPXGT means. Explain the abbreviation also in the abstract. Present each abbreviation (S. aureus…)
Staphylococcus aureus should be italic
Better explain in the introduction the term "ensemble docking". The introduction should explain why choosing this specific approach over other methods. Detail why other methods have failed in finding clinical inhibitors for sortaseA.
The manuscript rely on the work of Abujubara (16). It should clearly state what add new, and why the new work was important to be performed.
The discussion section could be improved by discussion other inhibitors of SrtA. See and reference the following review presenting a large collection of SrtA inhibitors and their chemical classification.
Chemical Biology of Sortase A Inhibition: A Gateway to Anti-infective Therapeutic Agents, Journal of Medicinal Chemistry, Vol 64/Issue 18
“Targeting Bacterial Sortases in Search of Anti-Virulence Therapies with Low Risk of Resistance Development”, Pharmaceuticals. 2021 Apr 30;14(5):415.
The methods protocol could be extended by detailing better the methods used. For example, better explain the choice of the dimensions of the grid box.
The discussion on the results should better detail the correlation or lack of it between the enzyme inhibition on the fret test and the results of the docking.
Author Response
Q1: Explain shortly what LPXGT means. Explain the abbreviation also in the abstract. Present each abbreviation (S. aureus…)
R1: All the abbreviations were explained at first use, thanks for pointing out.
Q2: Staphylococcus aureus should be italic
R2: We found all instances that should be italicized and fixed them
Q3: Better explain in the introduction the term "ensemble docking". The introduction should explain why choosing this specific approach over other methods. Detail why other methods have failed in finding clinical inhibitors for sortaseA.
R3: Initially we decided to briefly mention the "ensemble docking" term in the introduction in order to not overload a reader with terms, whereas the concept and its application are discussed in discussion section. However, if it's confusing for a reader, it's wise to broaden the initial description in Introduction. We made more explicit in the manuscript our reasoning about why the other methods are insufficient and why, we believe, clinical inhibitors are still lacking. Thank you for the suggestion. In short, we think that lead compound development requires reliable enough SBDD means. The latter is the main problem with such a target as SrtA having a flat and highly adaptable binding site. We propose a way to address this problem in the manuscript.
Q4: The manuscript rely on the work of Abujubara (16). It should clearly state what add new, and why the new work was important to be performed.
R4: As it was mentioned in the manuscript the experimental Abujubara work is well designed and provides interesting points for SAR. However, even all their finding could not be explained with the modeling they provide as well as with other similar modeling (we've made it a lot for SrtA and different ligands). Our work is a pilot (MVP) which helps to describe SAR Abujubara found in semi-qualitative manner for such a target being tough for modeling.
Q5: The discussion section could be improved by discussion other inhibitors of SrtA. See and reference the following review presenting a large collection of SrtA inhibitors and their chemical classification.
1. Chemical Biology of Sortase A Inhibition: A Gateway to Anti-infective Therapeutic Agents, Journal of Medicinal Chemistry, Vol 64/Issue 18
2. “Targeting Bacterial Sortases in Search of Anti-Virulence Therapies with Low Risk of Resistance Development”, Pharmaceuticals. 2021 Apr 30;14(5):415.
R5: Thank you for the suggestions and the references. We've read all the available literature on the topic. Our vision is that such a broad range of chemical classes only emphasizes the intrinsic problem of the target – it’s large and flexible. Provided none of the hits described in the provided references as well as other reviews focusing on chemical classes reached at least clinical candidate phase, it is clear that the main obstacle is not the incorrect chemical class, but rather the lacking means to develop hits to leads and drugs. It's the main focus of our research indeed.
We have added the proposed references (one of them was already cited in the manuscript) and several similar works in order to emphasize the contrast between the number and diversity of hit inhibitors and the lack of drugs/drug candidates/leads.
Q6: The methods protocol could be extended by detailing better the methods used. For example, better explain the choice of the dimensions of the grid box.
R6: We checked how to meaningfully expand the methods section and made several additions to enhance reproducibility of the results. The most notable addition considers the grid box, as you mentioned. We have added an explanation and the values of the crucial parameters, as well as included the pictures of the box places in all the SrtA conformations used in the work in Supplementary information.
Q7: The discussion on the results should better detail the correlation or lack of it between the enzyme inhibition on the fret test and the results of the docking.
R7: We believe that in detail comparison will unnecessary extend the text of the manuscript. Especially taking into account that the inconsistencies pertinent to docking inhibitors to SrtA are rather qualitative in the already available approaches using a single SrtA conformation. So we focused our attention on the qualitative improvements. Moreover in our pilot research we check the hypothesis that a wise choice of the MD generated conformations is a reasonable general approach for ensemble docking of such a tough target as SrtA. We believe a greater number of clusters could be necessary for a more production grade solution and we point it out in Discussion as one of the future directions.
Reviewer 2 Report
Comments and Suggestions for Authors
1. Abbreviate MRSA in line 34. Expanded form of abbreviations at first use must be accompanied by the parenthesized acronym.
2. Scientific names must be italicized. Throughout the manuscript, the authors need to change this.
3. In line 47, “Despite all the attractiveness of the SrtA drug target [8,10], no efficacious and developable drug candidates have been reported by the moment”. The language needs editing.
4. Line 52 “The site is rather shallow, thus resembling typical sites involved in protein-protein interactions whereas its two sides are defined by highly flexible loops” is confusing.
5. In silico in line 70 should be in italics
6. What is “shot sequences as LPRDA” in line 80.
7. The introduction is not structured well. The authors should introduce Sortase A (SrtA), the possibility of Sortase A (SrtA) as drug target discussing the structural aspects . What are the relevant observation from previous work? or how the current work is tackling any of the previous problems. What is the primary contribution or improvement to the topic study?
8. The lines 66-69 and 84-88 are confusing. Better to break it down to separate sentences to convey appropriate meaning.
9. Table 1 headings and labelling are confusing
10. The merits of this study are hard to ascertain due to the unclear language of the manuscript.
11. Results should only reflect the raw results from the study. Discussion should be shifted into the relevant section.
12. Materials and methods should not contain discussion! State the methods and steps in the order of execution along with conditions. List the tools used for each task. Short sentences will be sufficient or cite the relevant paper and shift details to supplementary.
Comments on the Quality of English LanguageThe manuscript needs extensive English language editing. The authors have merged several sentences together and it is extremely confusing. Authors must rewrite the manuscript to be considered for publication.
Author Response
Q1: Abbreviate MRSA in line 34. Expanded form of abbreviations at first use must be accompanied by the parenthesized acronym.
R1: We sought and expanded all abbreviations of the manuscript at first use.
Q2: Scientific names must be italicized. Throughout the manuscript, the authors need to change this.
R2: Thanks, we corrected all the terms which should be in italics.
Q3: In line 47, “Despite all the attractiveness of the SrtA drug target [8,10], no efficacious and developable drug candidates have been reported by the moment”. The language needs editing.
R3: The sentence was completely re-written to clearer covey the meaning.
Q4: Line 52 “The site is rather shallow, thus resembling typical sites involved in protein-protein interactions whereas its two sides are defined by highly flexible loops” is confusing
R4: The sentence was completely re-written to clearer covey the meaning.
Q5: In silico in line 70 should be in italics
R5: Thanks, we corrected all the terms which should be in italics.
Q6: What is “shot sequences as LPRDA” in line 80.
R6: Thanks, the typo was corrected.
Q7: The introduction is not structured well. The authors should introduce Sortase A (SrtA), the possibility of Sortase A (SrtA) as drug target discussing the structural aspects . What are the relevant observation from previous work? or how the current work is tackling any of the previous problems. What is the primary contribution or improvement to the topic study?
R7: The target has been extensively reviewed, so we decided to start from the problems which we and others pointed out previously. We formulate them briefly. The main idea of the introduction is to show that the adequate in silico means to develop hits/leads to drugs are still missing despite all the pharmacological attractiveness of the target. That is why we propose our approach to addressing the goal. It is essentially a pilot but the results are encouraging and worth further in detail development. We made this thought sequence more explicit in the Introduction.
Q8: The lines 66-69 and 84-88 are confusing. Better to break it down to separate sentences to convey appropriate meaning.
R8: We made clearer the sentences and the paragraphs including them.
Q9: Table 1 headings and labelling are confusing
R9: We clarified the headings and labeling of Table 1.
Q10: The merits of this study are hard to ascertain due to the unclear language of the manuscript.
R10: We undertook additional efforts to split long sentences and otherwise make the statements clearer. Hope it became much better now.
Q11: Results should only reflect the raw results from the study. Discussion should be shifted into the relevant section.
R11: We considered several options how to split the manuscript into sections. We have arrived at a structure in which the results are shortly discussed (in place), whereas the discussion section serves to place the results into a broader context and to outline the future directions we see starting from our pilot research.
Q12: Materials and methods should not contain discussion! State the methods and steps in the order of execution along with conditions. List the tools used for each task. Short sentences will be sufficient or cite the relevant paper and shift details to supplementary.
R12: We generally agree with the proposal, but finally decided to leave certain brief comments in order to help a reader to more properly interpret the choices made during the research. In case we remove these concise comments the risk of misunderstanding rises significantly. So we propose to leave this section in its current form (with certain clarifications requested by other reviewers).
Q13: The manuscript needs extensive English language editing. The authors have merged several sentences together and it is extremely confusing. Authors must rewrite the manuscript to be considered for publication.
R13: We undertook additional efforts to split long sentences and otherwise make the statements clearer. Hope it became much better now.
Reviewer 3 Report
Comments and Suggestions for Authors
The development of virtual study techniques such as AlphaFold, structure modeling, and docking has greatly elevated the status of structure-based drug design. The subject of this paper, Staphylococcus aureus, is one of the most notorious multidrug-resistant bacteria, and inhibitors utilizing the powerful class of peptidomimetics deserve significant attention. Therefore, I would like to propose a few suggestions or questions for the improvement of this well-written paper.
1. Elaborate on the specific reasons why SrtA was chosen among the many pathogenic proteins of Staphylococcus aureus.
2. What characteristics make certain types of proteins more suitable for research using structure-based drug design (SBDD)? Additionally, explain why other proteins may be more challenging to study using SBDD methods.
3. Despite the actual structure being available through PDB, what is the significant reasoning behind utilizing ensembles to predict the structure? Emphasize this further.
4. Is it possible to dock ligands such as glycans or carbohydrates with proteins? If so, which servers or tools would be most appropriate for this process?
5. As a final agenda, do you believe that modeled or predicted structures could one day fully replace actual experimental structures? Do you foresee a time when we will no longer need to determine unknown structures? Please explain why you think this is possible or not.
Author Response
Q1: Elaborate on the specific reasons why SrtA was chosen among the many pathogenic proteins of Staphylococcus aureus.
R1: The are several reasons, which were already provided in the previous version of Introduction. But we significantly rewrote the Introduction in order to clarify all the crucial points, so we believe that now that reasons became clearer.
Q2: What characteristics make certain types of proteins more suitable for research using structure-based drug design (SBDD)? Additionally, explain why other proteins may be more challenging to study using SBDD methods.
R2: That's a good point. The fact of sheer huge structural diversity of the published hits suggests that the target's site is not so uniquely defined. Generally when the site is well defined and rather small (as it is for many CNS targets) the ligand based design could be enough. When the site is large and rather flexible, SBDD can be used to at least choose a few viable options of binding among the greater number of possibilities, thus focusing the experimental research. Further on, if one of the binding hypotheses receives experimental validation, it can be used extensively for directed (rational) SBDD. A good example of a challenging SBDD targets is the already mentioned CNS (and other) receptors, whose very nature suggests a possibility of triggering between at least two states (more states are usually reported in detailed studies) - open and closed. It's not always that the precise open/close conformation of those receptors are available from either experiment or reliable modeling.But whether you are to develop an agonist or antagonist - the success will greatly depend on the available and reliable conformation of the receptor in each close/open form in SBDD settings.
Q3: Despite the actual structure being available through PDB, what is the significant reasoning behind utilizing ensembles to predict the structure? Emphasize this further.
R3: There are a few reasons for this.
1. as we and others discussed previously the PDB structures of SrtA raise several significant questions, thus are not as reliable as we may want
2. the conformational space of SrtA is large (again as we and other more elaborated MD simulations showed previously) and it is unlikely that the conformations presented in PDB are sufficient
3. if the available PDB conformations of SrtA were sufficiently reliable for SBDD, we would already have at least a few examples of hit2lead series made using SBDD
4. considering all the above we decide that using representative ensemble might be a good hypothesis to test further for such a pharmacologically promising target. Actually we did a pilot research in this direction in the manuscript
We significantly re-wrote the Introduction in order to clarify all the points we rely on, including the insufficiency of the existing PDB structures of SrtA.
Q4: Is it possible to dock ligands such as glycans or carbohydrates with proteins? If so, which servers or tools would be most appropriate for this process?
R4: Despite it's not the focus of our research, we have thoughts related to the topic.
1. the scoring functions used for docking are rather rough, so no special issue with interaction parameters should be expected
2. but there are two problems related to conformational sampling of the polysaccharides:
2.1 they could contain just too many freely rotatable (not in rings) single bonds, making the conformational search a tough problem for typical docking settings
2.2 even a more severe problem is that the cyclic sugar rings are rather flexible at body temperatures. The docking approachs split into two categories. In the first one, the ring part of a ligand is just kept fixed during the conformational sampling over the "freely rotatable bonds". In the second group, a set of representative conformations is generated for each ligand before the docking including its ring part. Then those conformations are used for rigid docking to the receptor. Both approaches fail for polysaccharides, since they were developed for drug-like molecules having in general far fewer accessible conformational degrees of freedom.
So we cannot just name an already established approach, however we heard that some people deal with this by wisely combining MD studies (for adequate sampling) with docking.
Q5: As a final agenda, do you believe that modeled or predicted structures could one day fully replace actual experimental structures? Do you foresee a time when we will no longer need to determine unknown structures? Please explain why you think this is possible or not.
R5: A good question. Personally I do not believe that the experimental structure determination would extinct one day. The nature has created very diverse patterns using the same building blocks. So I do believe there are two types of structures. First, the structures are representative of the well known and well statistically represented experimental structures. They are already and will be even better predicted well. It's the "interpolation case". On the other hand, there will be structures which (or the similar structures) are not statistically presented at the moment. They will be hard to predict correctly using ML/AI, etc. It's the "extrapolation case". For such cases more physics based relying methods are and will be used. It's hard to imaging that people would decide to spent so much money and effort to determine all the available patterns in protein structures in a foreseeable future, so the analysis provided, I think, will hold :)
Round 2
Reviewer 2 Report
Comments and Suggestions for Authors
The manuscript is revised well and can be accepted.
Try to break down any longer sentences for better readability for wider audience.
Comments on the Quality of English LanguageCheck the manuscript for possible typos and longer sentences.